# Estimating the population size of female sex workers and transgender women in Sri Lanka

**Ivana Bozicevic**[1]*, **Ariyaratne Manathunge**[2], **Zoran Dominkovic**[1], **Sriyakanthi Beneragama**[2], **Kelsi Kriitmaa**[3]

**1** World Health Organization Collaborating Centre for HIV Strategic Information, University of Zagreb School of Medicine, Zagreb, Croatia, **2** National STD/AIDS Control Programme, Ministry of Health, Colombo, Sri Lanka, **3** Philantrophy Advisory, Geneva, Switzerland

* Ivana.Bozicevic23@gmail.com

**Data Availability Statement:** All relevant data are within the manuscript and its Supporting Information files.

## Abstract

We implemented population size estimation of female sex workers (FSW) and transgender women (TGW) in Sri Lanka in 2018 using several approaches (geographical mapping, service and unique object multiplier and a modified Delphi method during the stakeholder consensus meeting). Mapping was done in 49 randomly selected Divisional Secretariats, which provided a basis for extrapolation of size estimates to the national level. Two types of adjustments were applied on the mean (minimum-maximum) population estimate obtained during mapping: (1) an adjustment for mobility to reduce double counting of FSW and TGW frequenting multiple spots, obtained during mapping; (2) an adjustment for "a hidden population", obtained from surveys among FSW and TGW. For the multiplier method, we used data from services of non-governmental organisations that FSW and TGW were in contact with, and surveys based on respondent-driven sampling. Surveys were carried out in the cities of Colombo (FSW, TGW), Kandy (FSW), Galle (FSW) and Jaffna (TGW). We estimated that there are 30,000 FSWs in Sri Lanka, with a plausible range of 20,000–35,000, which implies a prevalence of FSW of 0.56% (0.37–0.65%) among adult females. This study provided baseline estimates of 2,200 TGW in the country, with a plausible range of 2,000–3,500, which is 0.04% (0.04–0.07%) of adult male population. Our estimates of the proportional contribution of the FSW and TGW populations among the adult population in Sri Lanka are consistent with the The Joint United Nations Programme on HIV/AIDS (UNAIDS) recommended estimates for Asia and the Pacific. The results provide an important point for macro- and micro-level planning of HIV services, allocating programme resources and assessing programme coverage and quality.

## Introduction

Sri Lanka consists of nine provinces and 25 districts and has a population of 20.4 million [1]. According to the The Joint United Nations Programme on HIV/AIDS (UNAIDS) estimates, HIV prevalence among adults 15–49 years old is <0.1% [2]. An estimated 3500 [3000–4100] adults 15 years and above were living with HIV in Sri Lanka at the end of 2018 [3]. A total of

**Funding:** The study was funded by the New Funding Model Grant, 2016-2018 of the Global Fund to Fight AIDS, Tuberculosis and Malaria (https://www.theglobalfund.org/en/). The funders had no role in study design, data collection and analysis, decision to publish, or preparation of the manuscript.

**Competing interests:** The authors have declared that no competing interests exist.

350 HIV infected persons were newly reported during 2018, which is an increase of 23% compared to 2017. Sri Lanka has an HIV epidemic concentrated in key populations (KPs), primarily sex workers (SW), men who have sex with men (MSM), people who inject drugs (PWID), beach boys and transgender women (TGW). The term "beach boys" refers to young men who work near or on the beaches, typically tourist beaches, and who offer sexual services to men and women in exchange for some form of payment. They also include those working in restaurants, hotels, guest houses and boat-related tourism.

Knowing the size of KPs at higher risk of HIV is essential for intervention planning, resource allocation and for advocacy efforts. Measuring the size of KPs is, however, challenging since behaviours that put individuals at increased risk of HIV are often stigmatizing, which creates difficulties in reaching these populations.

The size of KPs can be estimated using different methods, each having its strengths and weaknesses [4, 5]. UNAIDS and WHO recommend that multiple methods are used in order to generate an estimate of a size of a KP for an area, given the variability in estimates produced by single studies [6]. Methods to estimate the size of KPs can be broadly categorized into methods based on collection of data directly from a KP (mapping, census and enumeration, multiplier, and capture–recapture), and methods whereby data are collected from a general population (population-based surveys, network scale-up) [7].

Geographical mapping carried out in Sri Lanka in 2013 estimated a total of 14,132 female sex workers (FSWs) (ranging from a minimum of 12,329 to a maximum of 15,935); 7,551 MSM (range 6,547–8,554), 17,459 (15,338–19,542) drug users (around 2% were estimated to inject drugs, which translates to approximately 350 PWID), and 1,314 (1,142–1,486) beach boys [8].

The aim of this paper is to describe results of the estimation of the population size of FSW and TGW in Sri Lanka using geographical mapping and the multiplier method. These two methods were deemed as the most appropriate approaches for two reasons: firstly, mapping was successfully done in 2013; secondly, in 2017–2018 Integrated Biological and Behavioural Surveillance (IBBS) surveys based on respondent-driven sampling (RDS) were carried out in FSW, PWID, MSM, transgender women (TGW) and beach boys, which provided an opportunity to implement the multiplier method in conjunction with the IBBS implementation. The choice of the geographical mapping and the multiplier method was also based on experiences from other countries in Asia where these methods have been used, and results applied to improve targeted HIV prevention and control programmes for KPs [9–13].

In Sri Lanka, as in the other countries in Asia, FSW and TGW are disproportionally affected by HIV. In the IBBS carried out in 2015, HIV prevalence among FSW in three cities in Sri Lanka ranged between 0.8–0.9% [3]. That round of IBBS did not include TGW as a separate category, a population that often faces social and legal exclusion, and is at an increased risk of experiencing violence and acquiring HIV [14, 15]. At the biological level, transgender women whose partners are males face a high HIV transmission probability via condomless anal sex.

## Materials and methods

Data collection methods included geographical mapping and the multiplier methods and finally a stakeholder consensus meeting, which provided an opportunity to discuss and interpret the results and agree on final KP size estimates.

FSW were defined as women who sold sex to men in exchange for money or goods in the past six months while TGW as persons who were assigned to be males at birth but who self-identify as transgender/transwomen and had penetrative anal sex with men in the past 12 months.

## Stage 1: Geographical mapping

The Divisional Secretariat division (DS division) was selected as the unit area for mapping, as this is the smallest administrative division in Sri Lanka that has defined boundaries. DSs were also selected as unit areas for mapping in the first round of geographical mapping carried out in Sri Lanka in 2013 [8]. Data on DSs were used from the Census of Population and Housing conducted in 2012 [1].

There is a total of 331 DSs in Sri Lanka, which we divided into three categories based on the population census 2012 information: high density (n = 110, total population ranging from 323,257–61,638), medium density (n = 110, 61,484–32,609 population) and low density (n = 111, 32,386–298 population) [16]. The Random Integer Set Generator was used to randomly select 25% (n = 28) of DSs from a high density category, 12.5% (n = 14) of DSs from a medium density category and 6.25% (n = 7) of DSs from a low density category [17]. Higher-density DSs were purposively over-sampled based on the assumption that they contain a greater concentration of KPs.

Data were collected through conducting Level 1 and Level 2 activities. Level 1 activities were used to collect information from secondary key informants (KIs) about locations where KPs socialize and about the minimum and maximum number of KP individuals that can be found at those locations. Secondary KIs include individuals who have close relationships with KPs, such as pimps, local food sellers, hotel staff, taxi drivers, and those that have professional knowledge about KPs such as health care staff, police and staff of non-governmental organisations (NGOs). KP members can also be included as secondary KIs. To expedite field work, selected DSs were divided into smaller areas based on administrative boundaries. Field supervisors collated data from Level 1 questionnaires on a daily basis in order to create unique lists of hotspots and estimates of the KP size.

Level 2 activities consisted of visiting hotspots from lists generated during Level 1 activities to validate that these were locations were KPs congregate, and interviewing KP members (also called primary KIs). During these interviews, information was collected about the minimum and maximum number of KP members that socialize at those hotspots on a peak day during a week and during an average month, typology of the hotspot (public place, brothel, night club, etc) and locations of other hotspots in the area. KP individuals interviewed at hotspots during Level 2 activities were also asked about their mobility across spots (in terms of how many hotspots on average they visit on a peak day). Estimates about population size obtained during Level 2 activities were taken as the final unadjusted estimates. To arrive at a single estimate, the mid-point ("mean") of the minimum and maximum estimates was used.

Questionnaires used to collect data during Level 1 and Level 2 activities were completed by trained field workers electronically, using tablets.

Data collection teams consisted of two interviewers representing non-KP and KP individuals to facilitate interaction with KIs, and a team co-ordinator.

The following adjustments were applied on the mean (minimum-maximum) estimate of the KP: (1) an adjustment for mobility to reduce double counting of FSW and TGW frequenting multiple spots, obtained during mapping; (2) an adjustment for "a hidden population", obtained from IBBS. IBBS questionnaires included questions to determine the proportion of respondents that do not visit outdoor places to find partners and socialize there with other members of their respective KP, and these were considered as "a hidden population".

The following formula was used to adjust for mobility:

$$S_2 = S_1(1 - p_1) + (S_1 \, p_1 \div m_1)$$

where $S_2$ was the estimated number of KP individuals adjusted for mobility, $S_1$ was the crude

number of KP individuals at a spot level obtained during Level 2 activities, $p_1$ was the proportion of KP individuals that visit more than one spot on a peak day, and $m_1$ was the mean number of spots that KP members visit in a peak day.

The following formula was used to adjust for a "hidden population":

$$S_3 = S_2 \div (1 - p_2)$$

where $S_2$ was the estimated size of a KP adjusted for mobility, $S_3$ was the estimated size after adjusting for "a hidden population", $p_2$ was estimated from IBBS, as a proportion of a KP who does not frequent hotspots to find partners and socialise with other members of their KP.

Adjusted KP size estimates were calculated for each spot. As a first step, all reported minimum and maximum KP estimates were summed up to get a range of estimate for each spot, which were summed up into DS estimates.

Further on, extrapolation was done to produce a point estimate and a range (low and high estimate) for the population size of FSW and TGW at a district, province and national level, using adjusted data. Firstly, a proportion of KP members among the total general population was calculated for each sampled DS. Secondly, a relationship between the estimated number of a KP and the general population was tested using the coefficient of determination ($R^2$) to assess plausibility of estimates. Thirdly, the pooled mean of proportions of mid-estimate, low and high estimate was calculated using the formula:

$$p = \frac{\sum_i^n M_{DS_i}}{\sum_i^n N_{DS_i}}$$

where p was the proportion of a KP among the general population, n was the number of sampled DSs (n = 49), $M_{DS}$ was the estimated number of a KP in a DS and $N_{DS}$ was the number of the general population in a DS based on the census data.

The pooled proportion was applied to census data of each DS, district, province and national counts to calculate population size estimates at the district, province and national level.

## Stage 2: Multiplier method

The multiplier method relies on two data sources that overlap in a known way: (1) data from an institution or service that KPs were in contact with, and (2) an IBBS representative of a target population. IBBS were carried out in the cities of Colombo (FSW, TGW), Kandy (FSW), Galle (FSW) and Jaffna (TGW).

We implemented the service and unique object multiplier method using four separate counts and the overlap of these counts with the IBBS. The service counts were: (1) number of FSW and TGW who were clients of the NGOs in each city in a defined three-month calendar period during the year 2017; (2) number of FSW and TGW who obtained condoms from these NGOs at least once in a defined three-month calendar period in 2017; (3) number of FSW and TGW that were escorted to STI clinics for STI screening by NGO staff in a defined three-month calendar period.

A fourth count was made through distribution of unique objects (purses) by NGO outreach workers to individual FSW and TGW at outreach venues in each city. Approximately 100 purses were distributed to KP members in the cities where IBBS were done two weeks before IBBS implementation. The number of unique objects distributed was determined by the number of contacts the distributing outreach organisations anticipated reaching within one week.

The number of venues in the cities where the unique object distribution occurred ranged between 11–32.

During administration of the survey questionnaires, all respondents were asked whether they received NGO services and purses by peer outreach workers during the specified time periods.

The choice of service multipliers was determined by the availability and quality of programme data. Eligibility criteria for the survey were consistent with those for NGO service utilization and for unique object distribution. The sampling area for IBBS was a city-level, and NGOs whose programme data were collected also operated at a city-level. Service data were extracted from the programmatic logbooks of NGOs that have unique codes assigned to individual clients and later on stored in an MS Excel file. Attention was paid that data are de-duplicated since these services could be obtained multiple times in a period of several months. During data analysis it emerged that data on the number of clients escorted to STI clinics by NGO staff were of low quality and therefore could not be used.

Following formula was used to estimate the size of FSW and TGW populations at a city-level: $N = n/p$, where N is the population size estimate, n is the count from the NGO programme registers and the number of unique object distributed, and p is the adjusted proportion of survey respondents reporting that they received a service or an object.

Methods and results of the IBBS are described elsewhere [18]. Briefly, participants were recruited using respondent-driven sampling (RDS), which is a quasi probability-based peer recruitment sampling method [19]. Recruitment started with a pre-determined number of seeds who were asked to recruit their peers from their personal social networks. The minimum sample size was determined to be 442 for FSW in Colombo, 307 in Galle and 341 in Kandy, while the following number of FSW were recruited in IBBS:: 458 in Colombo, 360 in Galle and 362 in Kandy. The minimum planned sample size of TGW was determined to be 250 in Jaffna and Colombo, respectively, while 254 TGW were recruited in Colombo and 252 in Jaffna. Overall, recruitment in IBBS was done from December 2017-March 2018.

Nine waves were reached for FSW in Colombo, seven waves in Galle and six waves in Kandy.

Survey data analysis was conducted in RDS-Analyst (RDS-A) software for univariate and bivariate analysis using Gile's SS (sequential sampler) estimator [20, 21]. This estimator takes into account network characteristics and generates adjusted proportions called "Estimated population proportions" with corresponding 95% confidence intervals. A 95% confidence interval (CI) of a single multiplier size estimate was calculated applying the 95% CI for a proportion of an indicator generated by the RDS-A. Three separate multiplier estimates for each city and population were synthesised using the median value, with the exception of data for Kandy [10, 22].

The use of several multipliers enables to obtain several size estimates and lowers the influence of bias had only one multiplier been used.

## Stage 3: Size estimation methods used after mapping and the IBBS surveys

Final population size estimates were reviewed and agreed upon at the community and stakeholder consensus meeting held in June 2018 using a modified Delphi method [23].

The aim of the consensus meeting was to interpret the results and establish upper and lower plausibility bounds for the KP size estimates. Stakeholders included representatives from the National STD/AIDS Control Programme of the Ministry of Health, research institutions, NGO representatives and KP community advocates.

At the beginning of the workshop, following the discussion of the study results, each participant provided his/her own estimates using data collection forms that were made available on Google Forms. These forms included questions on the estimated most likely, minimum (lower

bound) and maximum (upper bound) number of KPs in Sri Lanka. Participants were asked to classify each district into low/medium/high according to the estimated proportion of KPs among adult general population. Median values of the most likely, minimum and maximum estimates were used for the consensus estimate. In order to classify districts, each density category was assigned a value: low = 1, medium = 2, high = 3; each district was assigned an average density based on an arithmetic mean. If needed, adjustments to the estimates were made and additional rounds of estimations were conducted.

We also compared our estimates to the UNAIDS-recommended estimates for Asia and the Pacific, as a check on consistency of our estimates, and calculated the proportion of adult women and men in the general population who are FSW and TGW, respectively, using the census data [24].

## Ethical considerations

The protocols for the population size estimation and IBBS were submitted for the ethical approval to the Medical Faculty of the University of Sri Jayewardenepura, and the approval was obtained in October 2017.

Respondents provided an informed oral consent before participating in the study. Oral consent was witnessed by a team member and documented in an informed consent form.

## Results

### Female sex workers

For FSW, a total of 456 spots were validated during Level 2 activities, and 376 primary and 233 secondary KIs were interviewed. Mapping resulted in an estimate of 2,811 FSW (range from a minimum of 2,370 to a maximum of 3,251) at hotspots during an average month. -Spots were described as follows: street (39.5%), spa (20.8%), lodge/hotel (13.8%), shanti (small houses, usually made from pieces of wood, metal, or cardboard, in which poor people live, especially on the edge of a city; 9.6%), home (9.2%), other (1.9%), brothel (1.8%), karaoke/night club (1.1%), beach (0.9%), park (0.9%), massage parlour (0.4%). The peak days of activity for FSWs were reported to be Friday and Saturday. The average spot size is 6 FSW per spot. At 20.6% of spots FSW reported that they visited more than one hotspot on a peak day. The percentage of FSW estimated to be "hidden" (those who do not seek clients in public places) from IBBS ranged from 47.7% in Kandy to 73.6% in Galle.

Adjustment for mobility gave an estimate of 2,563 (range: 2,244–3,056), while an additional adjustment for a hidden population an estimate of 6,139 (5,249–7,180). The extrapolated estimated number of FSW in Sri Lanka based on mapping is 31,748 (27,148–37,131). The relationship between the estimated number of FSW and the census population was estimated at $R^2$ = 0.449.

The median estimated size of FSW in the city of Colombo based on the multiplier method is 2155 (95% CI 1812–2660) while in Galle it is 1134 (95% CI 983–1342). In Kandy, it was not possible to calculate the service-based multiplier estimates because of poor quality of NGO data. The unique object multiplier in Kandy produced an estimate of 685 (95% CI 543–925) FSW. Higher estimates of FSW were obtained via mapping compared to the multiplier method: 3,625 (3,099–4,239) of FSW in Colombo, 1,658 (1,418–1,939) in Galle and 2,145 (1,834–2,508) in Kandy.

There were two rounds of Delphi estimations done to reach the consensus estimate for FSW at the national level. The group agreed on the national point estimate of 30,000 and acceptable lower and upper bounds of 20,000 and 35,000 FSW, respectively. This translates to

a population prevalence of 0.56% (0.37–0.65%) FSW among adult women aged 15–49 years in Sri Lanka.

## Transgender women

Fifty-five spots were TGW socialise and find partners were visited during mapping, and 78 KIs were interviewed (44 primary and 34 secondary KIs). These spots were described as streets (65.5%), shanti (12.7%), beach (5.5%), public toilet (5.5%), home (1.8%) and other (9.1%). Mapping yielded an unadjusted estimate of 189 (range: 154–224) TGW. From IBBS data collected, 42.3% of TGW in Colombo and 69.9% in Jaffna do not visit outdoor places to find partners, and these were considered as estimates of "a hidden population" for the purpose of adjustment.

At only two spots respondents reported that they visited more than one spot on a peak day, and that included a visit to only one other spot. In selected DSs, adjustment for mobility gave a slightly lower average estimate of 183 (range: 149–217), while adjustment for "a hidden population" an estimate of 331 (269–380).

The median of the estimated population size based on the multiplier data for Colombo is 531 (95% CI 467–614), while in Jaffna it is 117 (95% CI 110–126). Lower size estimates were obtained via mapping: 195 (159–224) TGW in Colombo and 49 (40–56) in Jaffna.

The extrapolation yielded an estimate of 1,711 (1,393–1,966) TGW in Sri Lanka. The relationship between the estimated number of TGW and the census population was strong estimated at $R^2 = 0.393$.

The final PSE reached after the first round of the Delphi process was 2,200 (lower plausibility bound of 2,000 –higher plausibility bound of 3,500) of TGW at the national-level, which corresponds to 0.04% (0.04–0.07%) of TGW in the male general population aged 15–49.

## Discussion

This is the first study that used a range of sources of data and employed multiple methods to estimate the size of FSW and TGW in Sri Lanka. The results provide an important point for macro- and micro-level planning of HIV services at national and local level, and for allocating programme resources, including determining the volume of services required, planning of outreach work at the community-level and assessing programme coverage and quality.

Overall, the size estimate for FSWs in this study was considerably higher (30,000; lower plausibility bound of 20,000—upper plausibility bound of 35,000) than that found in the previous round of size estimation in 2013 (14,132; lower plausibility bound of 12,329—upper plausibility bound of 15,935) most likely due to correction factors that we employed to avoid substantially under-counting FSW. As there was no previous study estimating the size of TGW, this study provides baseline estimates of 2,200 TGW in the country, with a plausible range of 2,000–3,500.

Our estimates suggest that 0.56% (0.37–0.65%) of adult females in Sri Lanka might be FSW, which is consistent with the UNAIDS estimate for Asia and the Pacific (median of 0.35%, 25th percentile of 0.18%- 75th percentile of 2.33%).

Published data on size estimates of FSW are still limited for the Asian countries. Similar to our results, a study based on the multiplier method in Myanmar found a population prevalence of FSW among adult women (aged 18–49 years) of 0.35% (0.32–0.40%) in Yangon and 0.77% (0.69–0.84%) in Mandalay (10). In the Philippines, by using various approaches including mapping and the multiplier, it was estimated that 0.28% (0.19–0.40%) of females aged 15–49 in the general population are FSW, and their estimated absolute number is 66,100 (45,600–

95,300) [25]. In Nepal, it was estimated via mapping that there is a minimum of 43,829 and a maximum of 54,207 FSW, representing 0.47–0.58% of the adult female population [26].

Our estimate of the proportional contribution of the TGW population among the adult male population in Sri Lanka is also in concordance with the UNADS recommended estimates for Asia and the Pacific of 0.02% (25[th] percentile of 0.02-75[th] percentile of 0.06%).

There is still insufficient body of literature on size estimates of the transgender population [27]. The national estimate for TGW in the Philippines in 2015 was 122,800 or about 0.50% (0.40%-0.75%) of males aged 15–49, which is higher compared to our findings [25]. In Nepal, a study based on mapping found that there is a minimum of 18,704 and a maximum of 24,216 TGW, contributing with 0.22–0.29% to the male adult population [26]. In seven cities in Cambodia in 2012, it was assessed using the capture-recapture method that there were 2,690 (95% CI 2,600–2,780) TGW, with the highest number in Phnom Penh (1380, 95% CI 1350–1410), and 115–425 in other cities [28].

Geographical mapping has been the commonest method used to estimate the size of KPs in Asia [9]. An advantage of our approach to mapping was the random selection of DSs, which provided a good basis for the national-level extrapolation procedure, as well as inclusion of urban and rural areas. The selected mapping data extrapolation method assumes a linear relationship between the total general population and the number of KPs in DSs, i.e. that the proportion of a KP is constant in all geographical areas in a country. This may not be true for areas where some geographical characteristics are associated with the number of KPs, such as the number of tourists and the size of the cities (more opportunities for FSW in larger cities and tourist areas to meet clients). A statistical test of the relationship was conducted for each KP, showing strong evidence of a linear relationship.

We recognize that the methods presented in this paper have a number of limitations. Methods based on mapping tend to underestimate hidden populations. To address this weakness of the method, community guides were part of the mapping teams as that enabled to ensure a better access to information about KPs as well as KP themselves. However, FSW who contact clients through cell phones and social applications or are brothel or home-based, are likely to be under-represented. To address this challenge, final estimates were adjusted for a part of a population that is "hidden" i.e. unlikely to visit outdoor venues using data from IBBS. However, the quality of data on a sub-set of population that do not visit outdoor venues depends on the representativeness of IBBS.

The key source of a bias in the multiplier estimates is the selection bias leading to dependence between data sources, which could occur if those in contact with the NGOs were more likely to be included in the survey than those who were not NGO clients [29]. This could have been the case with the multiplier size estimates for FSW as they were considerably lower than the mapping estimates. Mapping produced lower estimates for TGW compared to the multiplier method which, as we hypothesise, could be due to difficulties with reaching TGW at outdoor venues. Efforts were taken to ensure that in both data sources needed for the multiplier method populations were defined in the same way and that unique objects were not distributed to ineligible individuals. We used the median of several multiplier-based size estimates to arrive at the single estimate with the lower and upper boundaries since this provides more robust data, not influenced by extreme values. Our estimates based on the multiplier method could have been more robust had more service-based data been available. This implies the need to improve data collection sources within HIV service delivery settings so that they can be used to develop multiplier size estimates in conjunction with future surveillance studies. For example, HIV testing data could not have been used for this round of size estimation as there are no individual-level testing records.

The Delphi method has also its limitations since it depends on the knowledge of participants about a certain KP, the type of work they are involved in and their level of expertise. The Delphi panel was composed of national experts who work on HIV programmes in key populations and efforts were made to ensure that the Delphi process in transparent and that each member of the panel was informed of the results of extrapolation of size estimates data to the national level and of the results of each round of the Delphi process. However, if the perception of Delphi panellists is far from the true number of key population individuals, the consensus estimates reached will reflect those biases.

This study provided an opportunity to enhance partnerships between governmental agencies and community-based organisations that provide services to KPs. To ensure an effective delivery of community-based prevention and linkage to HIV care interventions, legal and psychological services, efforts are needed to improve in particular the organizational, technical and monitoring and evaluation capacity of NGOs working with transgender communities. It is equally important to strengthen NGO's ability to access sub-groups of KPs who are not accessible at venues, and, as emerged from IBBS data, their numbers are considerable. This requires understanding of characteristics of FSW and TGW who are not socialising and meeting their clients/partners at outdoor venues but exclusively by other means (eg. through internet websites and mobile phone–based social networking apps) so that effective, targeted interventions can be developed. In relation to that, the next round of the population size estimation should consider utilising social media platforms to improve the coverage of various sub-sets of KPs.

## Conclusion

Overall, the approaches used in this round of population size estimation were robust and in line with the WHO and UNAIDS recommendations. Resulting estimates of the size of FSW and TGW should be used to improve the scale, coverage, and further roll-out of HIV interventions and evaluation of HIV prevention programmes among FSW and TGW.

## Supporting information

**S1 File. Multiplier-based population size estimates.**
(DOCX)

**S2 File. FSW data.**
(XLSX)

**S3 File. TGW data.**
(XLSX)

## Acknowledgments

We would like to thank the study participants and the staff of the National STD/AIDS Control Programme in Sri Lanka, Management Frontiers Ltd., and the staff from the following NGOs that participated in data collection: Abhimani, Saviya Development, Laksetha Sahana Sewa, Heart to Heart, Journey for Healthy Life, Family Planning Association, Wayamba Govi Sanwardhana Padanama, Rajarata Gami Pahana, Saviya Development Foundation, Natural Resource Development Foundation and

Sri Lanka Human Development Foundation.

The study was funded by the New Funding Model Grant, 2016–2018 of the Global Fund to Fight AIDS, Tuberculosis and Malaria (https://www.theglobalfund.org/en/).

The funders had no role in study design, data collection and analysis, decision to publish, or preparation of the manuscript.

## Author Contributions

**Conceptualization:** Ivana Bozicevic, Zoran Dominkovic.

**Formal analysis:** Ivana Bozicevic, Zoran Dominkovic.

**Funding acquisition:** Ariyaratne Manathunge, Sriyakanthi Beneragama.

**Methodology:** Ivana Bozicevic, Zoran Dominkovic.

**Project administration:** Ariyaratne Manathunge, Sriyakanthi Beneragama, Kelsi Kriitmaa.

**Resources:** Ariyaratne Manathunge.

**Supervision:** Ariyaratne Manathunge, Sriyakanthi Beneragama, Kelsi Kriitmaa.

**Validation:** Ivana Bozicevic, Ariyaratne Manathunge, Sriyakanthi Beneragama, Kelsi Kriitmaa.

**Writing – original draft:** Ivana Bozicevic.

**Writing – review & editing:** Ivana Bozicevic, Ariyaratne Manathunge, Zoran Dominkovic, Sriyakanthi Beneragama, Kelsi Kriitmaa.

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
