## [Decision Letter · Decision Letter 0]

7 Oct 2019

PONE-D-19-24615

Estimating the Population Size of Female Sex Workers and Transgender Women in Sri Lanka

PLOS ONE

Dear Ms Bozicevic,

Thank you for submitting your manuscript to PLOS ONE. After careful consideration, we feel that it has merit but does not fully meet PLOS ONE’s publication criteria as it currently stands. Therefore, we invite you to submit a revised version of the manuscript that addresses the points raised during the review process.

We would appreciate receiving your revised manuscript by 4th Nov 2019. To enhance the reproducibility of your results, we recommend that if applicable you deposit your laboratory protocols in protocols.io, where a protocol can be assigned its own identifier (DOI) such that it can be cited independently in the future. For instructions see: http://journals.plos.org/plosone/s/submission-guidelines#loc-laboratory-protocols

We look forward to receiving your revised manuscript.

Kind regards,

Kwasi Torpey, MD PhD MPH

Academic Editor

PLOS ONE

**Journal Requirements:**

2. Please include additional information regarding the survey or questionnaire used in the study and ensure that you have provided sufficient details that others could replicate the analyses. For instance, if you developed a questionnaire as part of this study and it is not under a copyright more restrictive than CC-BY, please include a copy, in both the original language and English, as Supporting Information. Moreover, please include more details on how the questionnaire was pre-tested, and whether it was validated.

3. As we note that data collection was performed by a cooperative group, we would recommend that you consult our authorship requirements page: https://journals.plos.org/plosone/s/authorship, to ensure that everyone who meets our criteria for authorship  is listed as an author.

4. Please provide additional details regarding participant consent. In the ethics statement in the Methods and online submission information, please ensure that you have specified (1) whether consent was informed and (2) what type you obtained (for instance, written or verbal, and if verbal, how it was documented and witnessed). If your study included minors, state whether you obtained consent from parents or guardians. If the need for consent was waived

5. When reporting the methodology of your expert opinion study, please ensure that the method of recruitment and selection of the experts, and their characteristics, are clearly explained so that the study could be reproduced.

**Comments to the Author**

1. Is the manuscript technically sound, and do the data support the conclusions?

Reviewer #1: Partly

Reviewer #2: Yes

Reviewer #3: Yes

2. Has the statistical analysis been performed appropriately and rigorously? 

Reviewer #1: No

Reviewer #2: Yes

Reviewer #3: Yes

3. Have the authors made all data underlying the findings in their manuscript fully available?

Reviewer #1: Yes

Reviewer #2: No

Reviewer #3: Yes

4. Is the manuscript presented in an intelligible fashion and written in standard English?

Reviewer #1: Yes

Reviewer #2: Yes

Reviewer #3: Yes

5. Review Comments to the Author

Reviewer #1: This manuscript reports population size estimates for female sex workers and transgender women in Sri Lanka. However, those estimates are not based upon a sound statistical analysis, which undermines their potential value. Further, this manuscript does not contribute to the advancement of methods for estimation of the sizes of key populations. I urge the authors to consult with a professional statistician before proceeding further with this manuscript

Any revision of this manuscript should address the following issues:

1. The “service multiplier” method is a redundant invention of epidemiology. It is mathematically identical to capture-recapture estimation, and should be treated as such. There is well-established literature and a rich set of estimation methods which should be used.

2. Each individual estimate for each service count should be reported, along with proper confidence intervals or Bayesian credible sets. It is not clear how “confidence intervals” were estimated for the medians of those estimates.

3. Evaluate whether your response data will support three- or four-source capture-recapture estimation from the service lists. The data will likely be adequate if each and every person contacted was asked about their use of each of the services. Multiple-source estimates will be more precise than multiple two-source estimates.

4. (Minor) Please provide a reference for the “Random Set Generator”.

5. The midpoint (average) of the minimum and maximum estimates is not the mean population size, and use of the word “mean” is inappropriate.

6. For adjustment for those who did not use hotspots, how was p2 estimated? I suspect it was estimated from the RDS, but readers should not need to guess.

7. How do you justify the use of the adjective “strong” for R2 = 0.449? Less than half of the variance in KP size is explained by population size. This is a weak measure of the success of the estimation.

Reviewer #2: Review: PONE-D-19-24615

Estimating the Population Size of Female Sex Workers and Transgender Women in Sri Lanka

Introduction

1. “In the latest IBBS carried out in 2015…” But further up you write “in 2017-2018 Integrated Biological and Behavioural Surveillance (IBBS) surveys”? Please clarify in the text when the latest surveys were carried out.

Methods

2. “In the absence of a gold standard method for estimating the size of a KP, it is recommended to implement multiple methods simultaneously to minimise potential bias resulting from a single method [6].” Suggest deleting – this is not pertinent to Methods and you already state similar in the Introduction.

3. “We also compared our estimates to the UNAIDS-recommended estimates for Asia and the Pacific, as a check on validity of our estimates” Being close to UNAIDS estimates does not make them valid. UNAIDS aggregate estimates are based on previous reported estimates and as such may be subject to similar biases as your or any other PSE activity. Rather, UNAIDS estimates may be useful to assess consistency of your estimates against regional UNAIDS estimates.

4. Please provide more detail about the unique object multiplier: How many days / weeks before the UO distribution? How many venues? How many objects?

5. Please provide more detail re the RDS: Sampling duration? Sample size? No. waves reached? Was convergence reached for receipt of the unique object? Were there bottlenecks re the unique object receipt? Similar for the NGO service / membership multiplier. Were eligibility criteria for the survey consistent with those for NGO service utilization / NGO membership / and UO receipt? Was the sampling area consistent? If you prefer describe this in Results.

Results

6. “shanti (9.6%),” Please explain what a Shanti is.

7. “Mobility was found at 20.6% of spots.” I don’t quite understand what that means, please rephrase?

8. “The relationship between the estimated number of FSW and the census population was strong (R2=0.449).” If you are using “strong” as a recognized category to interpret R2 values, please provide a reference. If this is your own opinion, just state the value and discuss the strength of the relationship in Discussion. (Same comment applies to the similar sentence for the TGW estimates: “The relationship between the estimated number of TGW and the census population was strong (R2=0.393).”)

9. “Adjustment for mobility gave a slightly lower average estimate of 183 (range: 149-217), while adjustment for “a hidden population” an estimate of 331 (269-380).” I’m not clear to which locality this estimate refers to – please clarify in the text?

10. If I read the text right you report the median of the various RDS based multiplier estimates? Please state all estimates, including the resulting median value.

Discussion

11. “The results provide an important point for macro- and micro-level planning of HIV services” Unclear?

12. “In seven cities in Cambodia in 2012, it was assessed using the capture-recapture method that there were 2,690 (95% CI 2,600 - 2,780) TGW, with the highest number in Phnom Penh (1380, 95% CI 1350-1410), and 115-425 in other cities [28].” Please add the relative PSE for these values.

13. Please discuss in greater detail the (very) large proportions of “hidden” KP, i.e. KP not attending venues. Do other studies report similarly large proportions?

Additional comments:

- I would not think that enumeration was used here which implies that you actually counted KP members at a subset of DS? Rather it seems you use key informants to give you’re their best guess. I suggest rephrasing methods description and terms used.

- While very true that no gold standard exists for PSE, the use of multiple methods does not mean that the resulting PSE necessarily improve in validity (that would only be the case if the various PSE are scattered both above and below the true PS value). If e.g., all methods used provide biased estimates what is the advantage of using multiple methods? I suggest discussing the merit of your approach in your Discussion section.

Reviewer #3: This is an important and interesting study focusing on estimating the population size of FSW and TGW in Sri Lanka. While the multiplier method is well-received and widely used method for estimating population size of KPs, geographical mapping with enumeration has a lot of weaknesses and challenges as they do not always meet the statistical rigor. A few queries: why having penetrative anal sex with was considered to as a necessary inclusion criteria. Would be good to have some clarification. The authors selected higher proportion of DSs from high density category. Would not that generate some bias. I was not sure if that was taken into consideration when analyzing the data to estimate at the national level. Would be good to have some clarification. The authors used the Delphi method to arrive at national and provincial estimates. Delphi method can always be biased because of the biases and interest of those at the meeting. Not sure if these are addressed.

6. PLOS authors have the option to publish the peer review history of their article (what does this mean?). If published, this will include your full peer review and any attached files.

Reviewer #1: No

Reviewer #2: No

Reviewer #3: No

---

## [Author Response · Author response to Decision Letter 0]

23 Dec 2019

Dear Reviewers, 

thank you for your comments, we appreciate a lot your work which helped us to improve the manuscript. The responses to your comments are provided as a separate file.

---

## [Editor Report · Decision Letter 1]

27 Dec 2019

Estimating the Population Size of Female Sex Workers and Transgender Women in Sri Lanka

PONE-D-19-24615R1

Dear Dr. Bozicevic,

We are pleased to inform you that your manuscript has been judged scientifically suitable for publication and will be formally accepted for publication once it complies with all outstanding technical requirements.

With kind regards,

Kwasi Torpey, MD PhD MPH

Academic Editor

PLOS ONE
---

## [Editor Report · Acceptance letter]

8 Jan 2020

PONE-D-19-24615R1 

Estimating the Population Size of Female Sex Workers and Transgender Women in Sri Lanka 

Dear Dr. Bozicevic:

I am pleased to inform you that your manuscript has been deemed suitable for publication in PLOS ONE. Congratulations! Your manuscript is now with our production department. 

With kind regards,

on behalf of

Professor Kwasi Torpey 

Academic Editor

PLOS ONE